# QuRe: Query-Relevant Retrieval through Hard Negative Sampling in Composed Image Retrieval

**Jaehyun Kwak** [1]   **Ramahdani Muhammad Izaaz Inhar** [1]   **Se-Young Yun** [1]   **Sung-Ju Lee** [1]

## Abstract

Composed Image Retrieval (CIR) retrieves relevant images based on a reference image and accompanying text describing desired modifications. However, existing CIR methods only focus on retrieving the target image and disregard the relevance of other images. This limitation arises because most methods employing contrastive learning–which treats the target image as positive and all other images in the batch as negatives–can inadvertently include false negatives. This may result in retrieving irrelevant images, reducing user satisfaction even when the target image is retrieved. To address this issue, we propose **Qu**ery-**Re**levant Retrieval through Hard Negative Sampling (QuRe), which optimizes a reward model objective to reduce false negatives. Additionally, we introduce a hard negative sampling strategy that selects images positioned between two steep drops in relevance scores following the target image, to effectively filter false negatives. In order to evaluate CIR models on their alignment with human satisfaction, we create Human-Preference FashionIQ (HP-FashionIQ), a new dataset that explicitly captures user preferences beyond target retrieval. Extensive experiments demonstrate that QuRe achieves state-of-the-art performance on FashionIQ and CIRR datasets while exhibiting the strongest alignment with human preferences on the HP-FashionIQ dataset. The source code is available at https://github.com/jackwaky/QuRe.

## 1. Introduction

Composed Image Retrieval (CIR) retrieves images from a large corpus using text and image inputs, enabling precise

[1]KAIST. Correspondence to: Sung-Ju Lee <profsj@kaist.ac.kr>.

*Proceedings of the 42nd International Conference on Machine Learning*, Vancouver, Canada. PMLR 267, 2025. Copyright 2025 by the author(s).

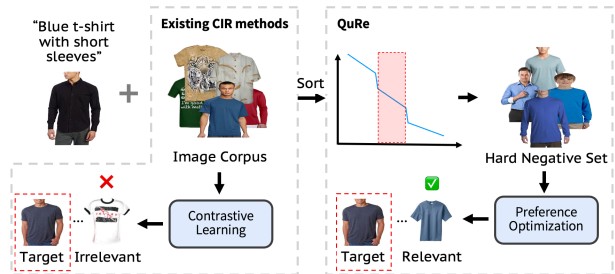

*Figure 1.* Comparison of existing CIR methods and QuRe. Traditional CIR approaches treat all non-target images as negatives in contrastive learning. In contrast, QuRe ranks the image corpus using a learned relevance score to identify a hard negative set. It then applies preference-based optimization to distinguish not only the target image but also other relevant images from hard negatives, leading to improved retrieval performance.

search capabilities in scenarios where textual descriptions alone are insufficient. This task is critical for applications such as e-commerce and Internet search, where users often seek results that match complex, visually nuanced specifications. For instance, as shown in Figure 1, a query combining an image of a shirt with the text 'blue t-shirt with short sleeves' requires CIR to retrieve matching images, including variations in style or color.

Existing CIR methods focus on retrieving the target image and overlook the broader relevance of other images. This limitation arises from the structure of CIR datasets, which typically annotate only a single target image per query and lack annotations for false negatives–relevant images not marked as targets. Furthermore, most CIR methods (Bai et al., 2024; Li et al.; Zhang et al., 2024) adopt contrastive learning, treating the target image as positive and all other images in the batch as negatives, inevitably including false negatives. While effective at ranking the target image within the top-k results, this approach frequently retrieves irrelevant images, as illustrated in Figure 1. Such irrelevance can reduce user satisfaction, as retrieval experience largely depends on the proportion of relevant items in the retrieved set (Al-Maskari & Sanderson, 2010).

We propose **Qu**ery-**Re**levant Retrieval through Hard Negative Sampling (QuRe), which aims to retrieve not only

the target image but also other relevant images with high ranks, thereby improving user satisfaction. To mitigate the inclusion of multiple false negatives, QuRE adopts a reward model training objective (Ouyang et al., 2022), optimizing the likelihood of ranking a positive image above a *single* negative image, with the target image designated as positive for each query. A key challenge lies in sampling appropriate negatives, excluding false negatives while incorporating *hard negative*.

Hard negatives are generally defined as samples that (1) belong to a different class than the anchor and (2) have embeddings close to the anchor (Robinson et al., 2020; Ma et al., 2020; Tabassum et al., 2022; Huynh et al., 2022). Traditional hard negative selection relies on class labels, but in CIR, each query has a unique target, making class-based distinctions impractical. Moreover, randomly selecting negatives from the entire corpus or choosing those too similar to the target often leads to suboptimal training (Figure 4). To address these challenges, we redefine the first condition as 'less relevant to the query than the target,' ensuring that hard negatives differ from the query in at least one key attribute, such as color or shape.

To balance both conditions for selecting proper hard negatives in CIR, QuRE periodically sorts the images in the corpus based on their relevance scores, calculated using the training model. During training, with the target image marked as positive, visually similar images (false negatives) tend to rank near the top of the sorted list. The sorted images are then divided into three groups: (1) false negatives, including the target image, (2) hard negatives, and (3) easy negatives. Hard negatives are defined as images that fall between two sharp declines in relevance scores, which occur after the target image. These steep drops indicate significant changes in relevance (Xia et al., 2024), ensuring that hard negatives differ from the query in at least one key attribute (Figure 6). This distinction makes hard negatives particularly valuable as challenging examples for training.

Our approach achieves state-of-the-art performance on the FashionIQ and CIRR datasets. To further evaluate alignment with human preferences, we created the Human-Preference FashionIQ (HP-FashionIQ) dataset, where human annotations indicate preferences between two retrieved sets for a given query. Experiments on HP-FashionIQ demonstrate that QuRE achieves the best alignment with human preferences compared to baseline methods.

Our contributions are as follows:

- We propose QuRE, a CIR algorithm that retrieves not only the target image but also other relevant images with high ranks to enhance user satisfaction.

- We introduce a novel hard negative sampling strategy that identifies images between two sharp declines in relevance scores after the target image. It optimizes model training by leveraging highly challenging samples while ensuring they are less relevant to the query than the target.

- We achieve state-of-the-art performance on the FashionIQ and CIRR datasets, and demonstrate superior alignment with human preferences on the newly introduced Human-Preference FashionIQ (HP-FashionIQ) dataset.

## 2. Related Work

**Vision-language foundation model.** Vision-Language Models (VLMs) have gained attention for their ability to integrate multimodal data. Transformer-based architectures effectively handle both visual and language inputs (Li et al., 2019; Lu et al., 2019). Contrastive learning methods, which align visual and language modalities, have significantly improved performance in VLMs (Jia et al., 2021; Radford et al., 2021). New architectures combine features from both modalities. For instance, Flamingo (Alayrac et al., 2022) and BLIP (Li et al., 2022) use cross-attention, where visual hidden states from the vision encoder are inserted into cross-attention layers within the text encoder layers. BLIP-2 (Li et al., 2023) and QWEN (Bai et al., 2023) utilize pre-trained image and text encoders with learnable networks that bridge the gap between modalities.

QuRE fine-tunes BLIP-2, using its image and text encoders with the Q-former module to handle modality gaps. We chose BLIP-2 for its efficient combination of image and text processing, requiring minimal training of the Q-former module.

**Composed image retrieval.** The CIR task fetches images using multimodal input features. A common approach is feature fusion, where the reference image and text are jointly embedded and compared against embeddings of candidate images (Vo et al., 2019; Dodds et al., 2020; Liu et al., 2021; Baldrati et al., 2023). Bi-BLIP4CIR (Liu et al., 2024) trains the text encoder using bi-directional training to capture both text directions of a given relation. CASE (Levy et al., 2024) leverages BLIP (Li et al., 2022) cross-attention architecture to perform an early fusion between the modalities. Other approaches transform images into pseudo-word embeddings or sentence-level prompts for text-to-image retrieval (Liu et al., 2023; Saito et al., 2023; Bai et al., 2024). MGUR (Chen et al., 2024) introduces an uncertainty loss for coarse-grained retrieval, and SPN4CIR (Feng et al., 2024) proposes a data generation method to scale positive and negative samples using multimodal LLMs.

However, all previous works trained models using contrastive loss, treating the target image as positive and all other images in the batch as negatives. This approach risks

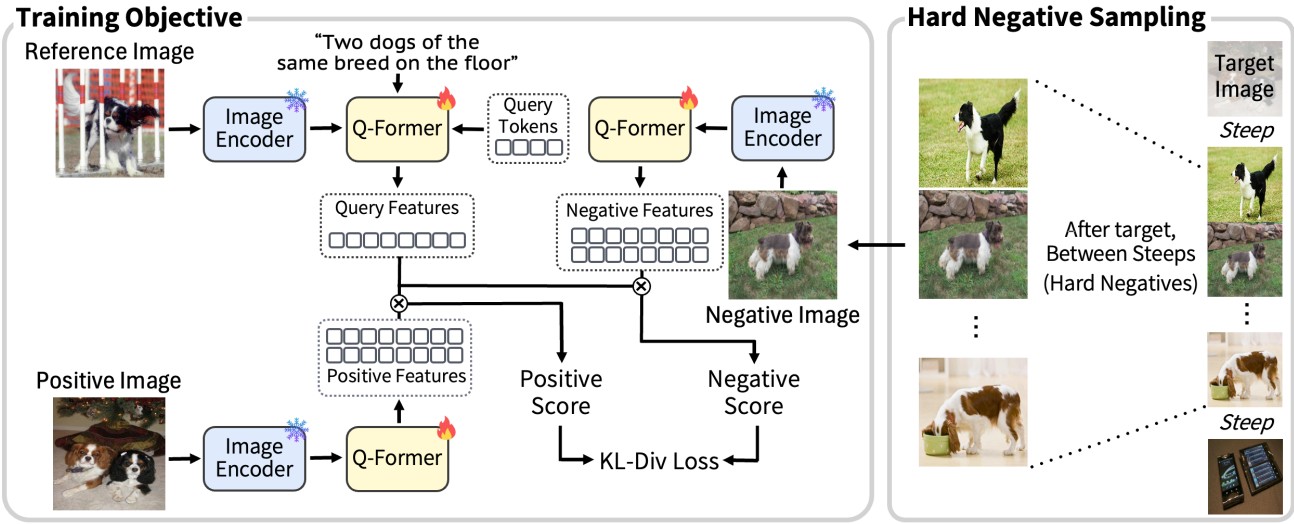

*Figure 2.* Overview of QuRe. During training, QuRe periodically ranks corpus images by relevance score using the current model. Hard negatives are selected from the range between two sharp drops in relevance scores following the target image. A KL divergence loss is then used to train the model to assign higher relevance to the positive image than to a randomly chosen hard negative.

including false negatives as negatives, which may lead to irrelevant retrieval results. QuRe overcomes this issue by employing the reward model objective, pairing each positive image with a sampled single hard negative.

**Hard negative sampling.** Hard negative sampling is a common technique in contrastive learning that selects more informative negatives rather than treating all images in the batch equally. HCL (Robinson et al., 2020) suggests that hard negatives should (1) belong to a different class than the anchor, and (2) have embeddings closer to it. However, in unsupervised settings like the CIR dataset, the lack of annotations makes it challenging to identify appropriate hard negatives. Embedding similarity alone can lead to including false negatives, prompting some approaches to incorporate model uncertainty, as higher uncertainty often indicates proximity to class boundaries (Ma et al., 2020). UnReMix (Tabassum et al., 2022) addresses this by combining embedding similarity with model uncertainty to identify suitable hard negatives. Meanwhile, FNC (Huynh et al., 2022) uses a predefined threshold to filter out false negatives and treats such samples as positives.

To address these challenges, QuRe introduces a novel hard negative sampling strategy. It selects hard negatives from images positioned between two sharp drops in relevance scores following the target image. The steep drop after the target ensures that the selected image differs from the query in at least one key attribute (e.g., color or shape), making it a suitable hard negative for training.

## 3. Methodology

We denote a CIR dataset as $\mathbb{D} = \{\boldsymbol{d}_i \mid i = 1, \ldots, N_{data}\}$, where each data point consists of a reference image, relative text, and a target image, i.e., $\boldsymbol{d}_i = \{x_{I_i}, x_{T_i}, y_{I_i}\}$. The goal of CIR is to retrieve a set of images from the image corpus $\mathbb{I} = \{I_j \mid j = 1, \ldots, N_{img}\}$, including the target image $y_I$, where the retrieved images reflect the specified relative text $x_T$ while preserving the visual properties of the reference image $x_I$. The training algorithm of QuRe is provided in Appendix A.

### 3.1. QuRe: Query-Relevant Retrieval through Hard Negative Sampling

**Relevance score.** For each image in corpus $I \in \mathbb{I}$, we define the relevance score to the query as the inner product of the bi-modal query and image embeddings:

$$s(x_I, x_T, I) = \frac{Q(E_{img}(x_I), x_T) \cdot Q(E_{img}(I))}{\tau}. \quad (1)$$

Here, $E_{img}$ is the BLIP-2 image encoder, $Q$ denotes the Q-Former, and $\tau$ is the learned BLIP-2 temperature. Q-Former processes the reference image embedding and relative text to align image and text modalities through cross-attention.

**Training objective.** QuRe is trained to maximize the probability of preferring the highly relevant image, *positive*, over the less relevant image, *negative*. We model the latent preference distribution $p^*$ using the Bradley-Terry model (Bradley & Terry, 1952), where $I_p$ and $I_n$ denote the positive and negative images, respectively.

$$p^*(I_p \succ I_n \mid x_I, x_T) = \sigma(s(x_I, x_T, I_p) - s(x_I, x_T, I_n)). \quad (2)$$

We set the target image as positive $I_p = y_I$. To include the negative image $I_n$, we construct the dataset $\mathbb{D}^* = \{(x_I, x_T, I_p, I_n) \mid (x_I, x_T, I_p) \in \mathbb{D}, I_n \in \mathbb{H}\}$, where $I_n$ is drawn from the hard negative set $\mathbb{H}$, defined in Section 3.2. The model is optimized by minimizing the negative log-likelihood (NLL) loss:

$$\mathcal{L} = -\mathbb{E}_{(x_I, x_T, I_p, I_n) \sim \mathbb{D}^*}[\log(p^*(I_p \succ I_n \mid x_I, x_T))]. \quad (3)$$

This objective is equivalent to minimizing the KL divergence between $p^*$ and a target distribution $p = [1, 0]$, ensuring the model prefers the positive image over the negative.

### 3.2. Hard Negative Set Sampling

**Defining hard negative set.** We defined the conditions for an appropriate negative image $I_n$ in the CIR setting as follows:

**C1.** The negative image $I_n$ should be less relevant to the query than the target image $I_p$.
**C2.** The relevance score of the negative image $I_n$ should be similar to that of the target image $I_p$.

However, identifying false negatives is practically infeasible as the CIR dataset annotates only the target image for each query. Additionally, selecting images that are overly dissimilar from the target may satisfy C1 but fail to meet C2, or vice versa, further complicating the selection process.

To balance these conditions, we sort the images in corpus $\mathbb{I}$ by their relevance scores obtained from the training model, forming an ordered set:

$$\mathbb{S}_i = \{s_{i,1}, \ldots, s_{i,N_{img}}\}, \quad \text{where } s_{i,1} \geq \cdots \geq s_{i,N_{img}} \quad (4)$$

where $\mathbb{S}_i$ represents the relevance scores sorted in descending order for the $i$-th query.

Based on the ordered set, we categorize images into three groups: (1) false negatives, which include the target image, (2) hard negatives, and (3) easy negatives. To identify hard negatives, we select **images positioned between two steep relevance score drops occurring after the target image**. The steep drop in relevance score indicates a noticeable shift in semantic similarity (Xia et al., 2024), helping to exclude false negatives while selecting negatives that are still challenging for the model. By focusing on this transition zone, we ensure that the selected negatives maintain a balance between similarity and distinction from the target image.

The subset of scores lower than the target score is defined as:

$$\mathbb{S}_i^{<targ} = \{s_{i,j} \mid s_{i,j} < s(x_{I_i}, x_{T_i}, y_i)\}. \quad (5)$$

From this subset, the indices of the top two largest degradations are identified as:

$$k_1, k_2 = \arg \text{top-2}_j (s_{i,j} - s_{i,j+1} \mid s_{i,j} \in \mathbb{S}_i^{<targ}). \quad (6)$$

Finally, the hard negative set for the $i$-th query is defined as:

$$\mathbb{H}_i = \{I_j \mid j \in [\min(k_1, k_2) + 1, \max(k_1, k_2)],$$
$$s_{i,j} < s(x_{I_i}, x_{T_i}, y_i)\}. \quad (7)$$

**Sampling hard negatives.** To ensure diverse and informative negatives, a single image is sampled from the defined hard negative set for every epoch based on a uniform distribution.

## 4. HP-FashionIQ Dataset

The commonly used evaluation metric, Recall@k, fails to capture user satisfaction. While user satisfaction increases with the number of relevant items retrieved (Al-Maskari & Sanderson, 2010), Recall@k only checks whether the target image is retrieved, disregarding the relevance of other images in the fetched result. However, assessing the relevance of retrieved images is challenging, as it requires evaluating how well the images align with both the text and image inputs, which in CIR involves considering numerous complex attributes.

Human evaluation remains the most reliable way to measure image relevance, as humans can accurately assess how well an image matches a multi-modal query. To facilitate such evaluation, we created the Human-Preference FashionIQ (HP-FashionIQ) dataset, using the validation set of the FashionIQ dataset (Wu et al., 2021) with 61 participants. We selected the FashionIQ dataset for its high relevance and broad applicability, mirroring the search functionalities of e-commerce platforms.

*Table 1.* Statistics of HP-FashionIQ Dataset: Each query has two sets of retrieved images from different CIR models, annotated based on their preferences.

| # Total Queries | # Shirts Queries | # Toptee Queries | # Valid Queries |
|---|---|---|---|
| 3,050 | 1,800 | 1,250 | 2,715 |

**Data collection setting.** Each question consisted of two retrieved image sets, each with the top 5 results from different CIR models. For every question, two CIR models were randomly selected from the following four: CLIP4CIR (Baldrati et al., 2023), Bi-BLIP4CIR (Liu et al., 2024), CoVR-BLIP (Ventura et al., 2024b), and SPRC (Bai et al., 2024). We provided queries and retrieved images from the 'shirts' or 'top tees' categories of the FashionIQ dataset to participants. Each participant was given 50 questions with a total of 100 sets of retrieved images, covering 3,050 queries in the FashionIQ validation set.

**Annotation methodology.** For each question in the survey, participants chose the preferred set between the two provided sets, assessing the alignment with human preferences.

*Table 2.* Performance comparison on the FashionIQ validation dataset across different methods. The best results are highlighted in bold, and the second-best are underlined.

| Method | Dress | | Shirt | | Toptee | | Average | | |
|---|---|---|---|---|---|---|---|---|---|
| | R@10 | R@50 | R@10 | R@50 | R@10 | R@50 | R@10 | R@50 | Avg. |
| CoSMo (Lee et al., 2021) | 23.60 | 49.18 | 18.11 | 43.18 | 24.63 | 54.31 | 22.11 | 48.89 | 35.50 |
| MGUR (Chen et al., 2024) | 23.15 | 48.74 | 18.99 | 43.47 | 25.55 | 52.83 | 22.56 | 48.35 | 35.46 |
| CLIP4CIR (Baldrati et al., 2023) | 38.32 | 63.90 | 44.31 | 65.41 | 47.27 | 70.98 | 43.30 | 66.76 | 55.03 |
| Bi-BLIP4CIR (Liu et al., 2024) | 39.12 | 62.92 | 39.21 | 62.81 | 44.37 | 67.06 | 40.90 | 64.26 | 52.58 |
| CoVR-BLIP (Ventura et al., 2024b) | 44.55 | 69.03 | 48.43 | 67.42 | 52.60 | 74.31 | 48.53 | 70.25 | 60.24 |
| SPRC (Bai et al., 2024) | 45.71 | **70.00** | 51.37 | 72.77 | 55.48 | 77.46 | 50.86 | 73.41 | 62.13 |
| QuRe | **46.80** | 69.81 | **53.53** | **72.87** | **57.47** | **77.77** | **52.60** | **73.48** | **63.04** |

To our knowledge, this is the first CIR dataset with human preference-annotated retrieved images. An example of data from HP-FashionIQ is shown in Figure 3, with a detailed explanation of the data collection process in Appendix C.

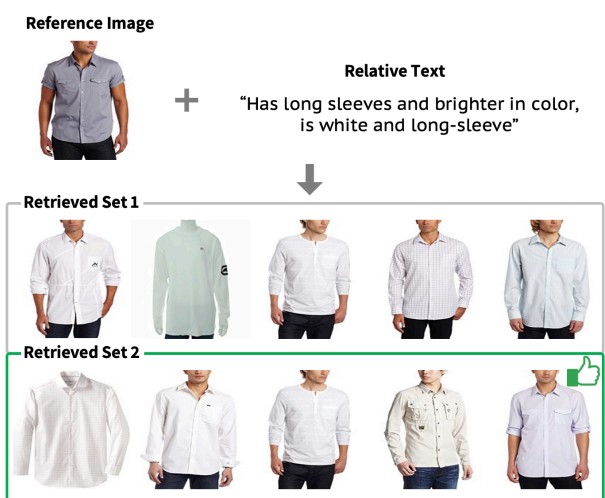

*Figure 3.* An example from the HP-FashionIQ dataset: Given a query, two retrieved image sets are presented. In this example, the user preferred Set 2, as it better preserves the visual properties of the reference image.

**Modality redundancy check.** While CIR should consider both image and text, some examples focus solely on the image or the text. CASE (Levy et al., 2024) highlighted modality redundancy in FashionIQ, indicating that text is sometimes more influential than the image. We instructed participants to consider both input modalities equally. We asked them to flag instances where one modality seemed irrelevant to the retrieved images to exclude data that was unclear for human evaluation. A total of 307 queries were treated as irrelevant and excluded.

**Sanity check.** To identify instances of decreased user con-

centration during the annotation period, users rated the relevance of the retrieved sets before annotating for their preferred choice. Users rated each set on a 5-point Likert scale (Likert, 1932), with a score of 5 indicating a strong match with the query. Queries were discarded if the user did not prefer the set with the higher relevance score. As a result, 28 queries were excluded, leaving 2,715 valid queries. The total number of queries in the HP-FashionIQ dataset is shown in Table 1.

## 5. Experiments

### 5.1. Experimental Setup

**Datasets.** We evaluate the models on widely used CIR datasets, FashionIQ (Wu et al., 2021) and CIRR (Suhr et al., 2018), to assess their ability to retrieve the target image. Additionally, we evaluate them on the HP-FashionIQ dataset to assess their alignment with human preferences.

**Implementation.** We used BLIP-2 (Li et al., 2023) with a ViT-L image encoder. Following previous work (Baldrati et al., 2023), we resized images to 224×224 with a 1.25 padding ratio. QuRe is trained using the AdamW optimizer (Loshchilov, 2017) for 50 epochs on CIRR and 30 epochs on FashionIQ. The hard negative set $\mathbb{H}$ was defined $n_{def}$ times, starting with a warm-up phase where $\mathbb{H}$ initially included the entire corpus except for the target during the first $\lfloor n_{epoch}/n_{def} \rfloor$ epochs. The hard negative set $\mathbb{H}$ is updated every $\lfloor n_{epoch}/n_{def} \rfloor$ epochs. We set $n_{def}$ to six for both FashionIQ and CIRR. All experiments were conducted using a single Nvidia RTX 3090 GPU.

### 5.2. Comparison with State-of-the-art CIR Models.

Table 7 presents the evaluation of CIR models on the FashionIQ dataset. QuRe consistently achieves the best or second-best performance across all categories, attaining the highest overall average. Notably, QuRe demonstrates

*Table 3.* Performance comparison on the CIRR test dataset across different methods, where Recall$_s$@K represents Recallsubset@K. The best results are highlighted in bold, and the second-best are underlined.

| Method | Recall@K | | | | Recall$_S$@K | | | Average |
|---|---|---|---|---|---|---|---|---|
| | K=1 | K=5 | K=10 | K=50 | K=1 | K=2 | K=3 | R@5 + R$_S$@1 |
| CosMo (Lee et al., 2021) | 6.48 | 23.11 | 34.63 | 67.33 | 20.29 | 40.22 | 60.80 | 43.55 |
| MGUR (Chen et al., 2024) | 5.78 | 21.45 | 33.42 | 67.06 | 20.29 | 40.22 | 60.80 | 42.91 |
| CLIP4CIR (Baldrati et al., 2023) | 44.12 | 77.23 | 86.51 | 97.95 | 73.11 | 89.11 | 95.42 | 75.17 |
| Bi-BLIP4CIR (Liu et al., 2024) | 32.55 | 64.36 | 76.53 | 91.61 | 63.54 | 82.46 | 92.48 | 63.95 |
| CoVR-BLIP (Ventura et al., 2024b) | 39.76 | 70.15 | 80.89 | 95.01 | 72.46 | 87.86 | 94.77 | 71.30 |
| SPRC (Bai et al., 2024) | 50.75 | 80.58 | 88.72 | 97.59 | **79.57** | **91.76** | **96.70** | 80.07 |
| **QURE** | **52.22** | **82.53** | **90.31** | **98.17** | 78.51 | 91.28 | 96.48 | **80.52** |

significant improvements over SPRC (Bai et al., 2024) when the retrieved set size is small, such as in Recall@10, with gains of 1.09%, 2.16%, and 1.99% for the dress, shirt, and toptee categories, respectively. Previous work (Levy et al., 2024) has identified high modality redundancy in the FashionIQ dataset, where text dominates the retrieval process, favoring text-based methods such as Bi-BLIP4CIR (Liu et al., 2024) and SPRC (Bai et al., 2024). Despite this bias, QURE achieves state-of-the-art performance, improving the overall average recall by 10.46% and 0.91% compared with Bi-BLIP4CIR and SPRC, respectively. These results highlight QURE's effectiveness in accurately retrieving the target image, even in the presence of modality redundancy.

Table 3 shows the evaluation results on CIRR, a general-domain dataset. QURE achieves the highest performance across all Recall@k metrics, particularly excelling in Recall@1 and Recall@5, surpassing the current state-of-the-art method, SPRC (Bai et al., 2024), by 1.47% and 1.95%, respectively. Regarding Recall$_s$@k, which measures retrieval performance from a subset containing relevant images and the target, QURE achieves the second-best results. This result is attributed to the design of QURE, where even false negative images can receive higher scores than the target as they closely match the query. This behavior arises from our hard negative set definition, which excludes false negatives, ensuring that relevant images are not treated as negatives and can be ranked higher than the target. Notably, QURE achieves state-of-the-art performance on the combined Recall@5 + Recall$_s$@1 average.

### 5.3. Evaluation with the HP-FashionIQ Dataset

We evaluate the alignment of CIR models with human preferences. Given a query $\{x_I, x_T\}$, each participant was presented with two different sets of retrieved images, $Set$ 1 and

$Set$ 2, and annotated their preferences between them. To calculate the overall relevance score of a CIR model for each set, we averaged the relevance scores of the five retrieved images within the set:

$$s_{rel}(Set\ i) = \frac{1}{5} \sum_{I \in Set\ i} s_{rel}(x_I, x_T, I). \quad (8)$$

where $s_{rel}$ denotes the relevance score (e.g., cosine similarity) computed by CIR models.

The alignment with human preferences is measured through the *preference rate*, which represents the conditional probability that $Set$ 1 is preferred when its relevance score is greater than that of $Set$ 2. Formally, we define the preference rate as:

$$\mathbb{P}(Set\ 1 \succ Set\ 2 \mid s_{rel}(Set\ 1) > s_{rel}(Set\ 2)). \quad (9)$$

Table 4 shows that the ranking of CIR models based on their alignment with human preferences on the HP-FashionIQ dataset differs from their performance rankings on the FashionIQ dataset (Table 7) using the Recall@k metric. For instance, MGUR achieves performance comparable to CosMo in retrieving the target image but aligns more closely with human preferences. This discrepancy stems from MGUR's additional coarse-grained loss, which considers both the target image and visually similar alternatives as positives. Moreover, while SPRC, CoVR-BLIP, and Bi-BLIP4CIR surpass CLIP4CIR in terms of Recall@k on the FashionIQ dataset, CLIP4CIR aligns better with human preferences on the HP-FashionIQ dataset, despite its lower accuracy in retrieving the exact target image.

QURE achieves the best alignment with human preferences, which shows that Set 1 is preferred 74.55% of the time when its relevance score exceeds that of Set 2. This result

*Table 4.* Preference rate comparison on HP-FashionIQ dataset across different methods. The best results are highlighted in bold, and the second-best are underlined.

| Method | Preference Rate (%) |
|---|---|
| CosMo (Lee et al., 2021) | 72.96 |
| MGUR (Chen et al., 2024) | 73.99 |
| CLIP4CIR (Baldrati et al., 2023) | 74.45 |
| Bi-BLIP4CIR (Liu et al., 2024) | 67.33 |
| CoVR-BLIP (Ventura et al., 2024b) | 73.15 |
| SPRC (Bai et al., 2024) | 73.82 |
| QuRe | **74.55** |

*Table 5.* Zero-shot performance comparison on the CIRCO dataset across different methods. The best results are highlighted in bold, and the second-best are underlined.

| Method | mAP@5 | mAP@10 | mAP@25 | mAP@50 |
|---|---|---|---|---|
| CosMo | 0.31 | 0.40 | 0.47 | 0.53 |
| MGUR | 0.14 | 0.17 | 0.25 | 0.30 |
| CLIP4CIR | 10.58 | 11.18 | 12.32 | 12.96 |
| Bi-BLIP4CIR | 4.74 | 4.97 | 5.69 | 6.10 |
| CoVR-BLIP | 18.35 | 19.25 | 21.02 | 21.88 |
| SPRC | 17.57 | 18.48 | 20.14 | 20.98 |
| QuRe | **23.22** | **24.23** | **26.26** | **27.24** |

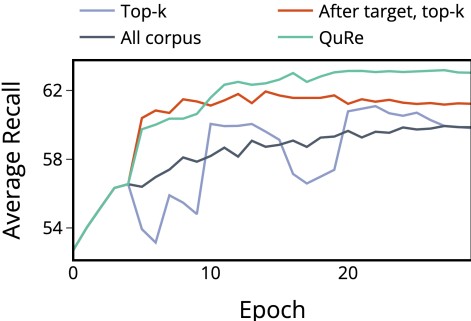

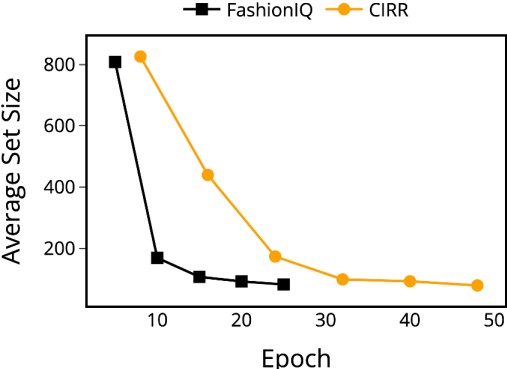

*Figure 4.* Average recall on the FashionIQ validation set using four hard negative set definitions. After the initial hard negative set is established (e.g., at epoch 4), the choice of definition significantly influences average recall throughout training.

*Figure 5.* Average size of the hard negative set per epoch on FashionIQ and CIRR. This plot shows how the average number of hard negatives per query evolves across epochs in which the set is defined.

demonstrates QuRe's ability not only to retrieve the correct target image but also relevant images that best align with human preferences.

### 5.4. Ablation Studies

We present ablation results under various scenarios, with additional experiments included in Appendix B.

**Zero-shot performance comparison.** We evaluated the zero-shot performance of the models on the CIRCO dataset using those pre-trained on the CIRR dataset. Although QuRe and baseline methods are not explicitly designed for zero-shot tasks, models that effectively retrieve relevant images are expected to perform well in such scenarios. Furthermore, CIRCO is the first CIR dataset to include multiple ground truths, addressing the issue of false negatives in existing datasets. Thus, evaluating the mean average precision at k (mAP@k) on this dataset provides a reliable measure of the model's ability to retrieve relevant items.

Table 5 reveals that QuRe achieves the best performance among all baselines, outperforming the second-best method, CoVR-BLIP, by an average margin of 5.13 mAP. While

Table 3 indicates SPRC (Bai et al., 2024) significantly outperforms CoVR-BLIP (Ventura et al., 2024b) on the CIRR dataset, CoVR-BLIP shows better performance in the zero-shot setting. This suggests that CoVR-BLIP generalizes unseen tasks better. The results highlight that QuRe achieves the highest generalizability, significantly improving over other baselines.

**Effect of hard negative set definition strategy.** To assess the effectiveness of the hard negative set definition approach, we compare four strategies on the FashionIQ dataset. The hard negative set is defined as follows: (1) the entire image corpus (*All corpus*), (2) the top-k images based on relevance scores (*Top-k*), (3) the top-k images with relevance scores lower than the target (*After target, top-k*), and (4) images between sharp drops in relevance scores, occurring after the target (QuRe).

Figure 4 presents the average recall of each strategy across training epochs. The results indicate that the *All corpus* approach shows consistent improvement; however, it ul-

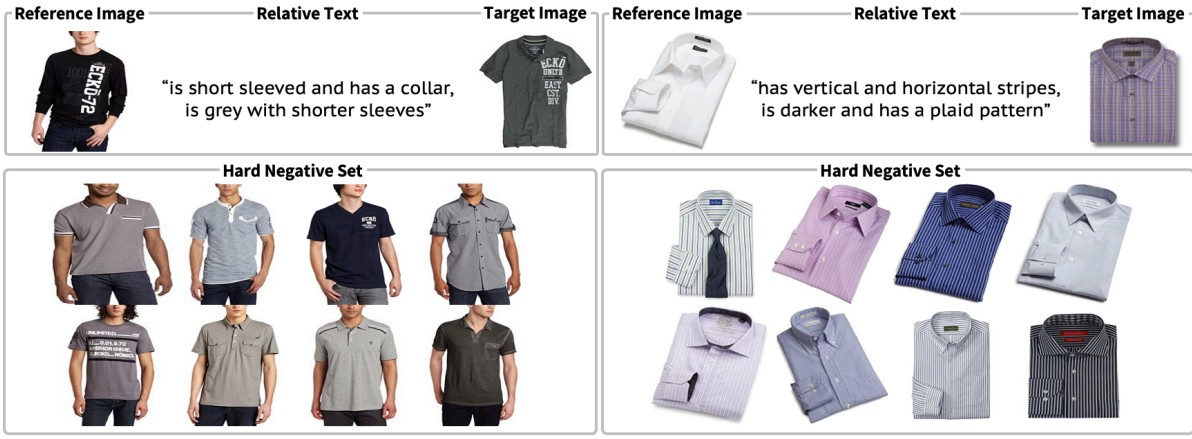

*Figure 6.* Hard negative set examples for two shirt-category queries in FashionIQ. Each set contains images that are semantically similar to the target but deemed less relevant by the model.

timately leads to suboptimal performance as the selected negatives from the entire corpus may be too irrelevant. Inspired by HCL (Robinson et al., 2020), we define a hard negative set by selecting the top-100 images based on relevance scores (*Top-k*). This approach occasionally enhances model performance (e.g., at epochs 10 and 20), but it often degrades due to the inclusion of false negatives. As training progresses, since the target image is treated as positive, visually similar images (false negatives) tend to rank higher. To mitigate the issue in the *Top-k* approach, we analyze the performance of the *After target, top-k* strategy, which excludes false negatives by selecting the top-k images following the target. Unlike *Top-k*, this method provides consistent improvements without fluctuations but gradually leads to slight performance degradation over time. This decline results from the heuristic assumption that all images right after the target are reliable hard negatives. In contrast, QᴜRᴇ identifies points where relevance drops sharply and selects images between two such points, resulting in stable performance improvements throughout training. These results show that, with a novel hard negative sampling method, QᴜRᴇ effectively balances the conditions C1 and C2 of selecting proper hard negatives.

**Size of hard negative set.** The number of hard negative images varies significantly depending on the complexity of bi-modal queries and the corpus. For instance, when the input text contains fewer attributes, such as 'blue shirt with short sleeves', the hard negative set may include images that match either 'blue' or 'short sleeves', resulting in a larger set. Therefore, it is crucial to define a query-specific hard negative set size (Xia et al., 2024).

Figure 5 shows the average size of the hard negative set across all queries in the FashionIQ and CIRR datasets. While QᴜRᴇ does not explicitly define the size of the hard negative set, the results indicate a consistent decrease

throughout training. Initially, the model identifies a broader range of images as hard negatives due to lower confidence. As training progresses, it refines this selection, yielding a smaller yet more challenging hard negative set. This dynamic resembles curriculum learning, where increasingly difficult samples accelerate model convergence.

**Visualization of hard negative set.** To evaluate our hard negative set sampling strategy, we present qualitative examples from the FashionIQ dataset in Figure 6, illustrating two queries. In the first query, the hard negatives either lack a collar or text from the reference image 'ECK' or depict shirts instead of t-shirts, differing from the input image. In the second query, all retrieved shirts contain only vertical stripes while successfully retrieving darker shirts. These examples demonstrate that the hard negative sets defined by our method consist of images that are less relevant than the target (C1) while remaining semantically similar (C2), thereby satisfying both conditions outlined in Section 3.2.

# 6. Conclusion

We present QᴜRᴇ, a CIR model designed to retrieve both target and relevant images to enhance user satisfaction. Existing CIR datasets typically annotate only the target image per query, leading prior methods to rely on contrastive learning that treats all non-target images as negatives. QᴜRᴇ mitigates false negatives by leveraging reward model objectives and introduces a novel hard negative sampling strategy, selecting images between two sharp relevance score drops after the target. To evaluate the alignment of CIR models with human preferences, we introduce the Human-Preference FashionIQ (HP-FashionIQ) dataset. QᴜRᴇ achieves state-of-the-art performance on both the FashionIQ and CIRR datasets and demonstrates the highest alignment with human preferences on the HP-FashionIQ dataset.

## Acknowledgements

This work was supported by the National Research Foundation of Korea (NRF) grant funded by the Korea government (MSIT)(RS-2024-00337007, 50%), Institute of Information & Communications Technology Planning & Evaluation (IITP) grant funded by the Korea government (MSIT) (RS-2019-II190075, Artificial Intelligence Graduate School Program (KAIST), 5%), and the Institute of Information & Communications Technology Planning & Evaluation (IITP) grant funded by the Korea government (MSIT) (No. 2022-0-00871, Development of AI Autonomy and Knowledge Enhancement for AI Agent Collaboration, 45%)

## Impact Statement

This paper introduces **Qu**ery-**Re**levant Retrieval through Hard Negative Sampling (QuRe), an approach to Composed Image Retrieval (CIR) that improves retrieval quality by retrieving both the target and other relevant images, improving user satisfaction. It addresses the false negative problem in CIR with a novel hard negative sampling strategy, advancing Machine Learning and Information Retrieval. QuRe contributes to improving search efficiency in e-commerce and visual search platforms, which benefits businesses and consumers by delivering more relevant search results. Additionally, the human preference-based evaluation (HP-FashionIQ dataset) aligns CIR models with human expectations, making AI-powered retrieval more user-centric.

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

# A. Algorithm

---

**Algorithm 1** Training Flow of QuRe

---

1: **Input:** Parameters $\theta$, CIR dataset $\mathbb{D}$, Image corpus $\mathbb{I}$, Number of defining negative set $n_{\text{def}}$, Total epochs $n_{\text{epoch}}$
2: **for** each epoch $e$ **do**
3:    **if** $e == 0$ **then**
4:       $\mathbb{H} \leftarrow \mathbb{I} \setminus y_I$                                                                          *// Warmup: using all candidates as the negative set*
5:    **else if** $e > 0$ **and** $e \bmod \lfloor n_{\text{epoch}}/n_{\text{def}} \rfloor == 0$ **then**
6:       $\mathbb{H} \leftarrow \{I \mid s(x_I, x_T, y_I) < s(x_I, x_T, I), I \in \text{range between two largest score degradations}\}$    *// Equation (7)*
7:    **end if**
8:    **for** each batch $b$ **do**
9:       $I_{n_b} \leftarrow \text{Sample}(\mathbb{H}_b)$                                                                *// Sample one negative for every query*
10:       $\mathcal{L} \leftarrow -\log(\sigma(s(x_{I_b}, x_{T_b}, I_{p_b}) - s(x_{I_b}, x_{T_b}, I_{n_b})))$                            *// Equation (3)*
11:       $\theta \leftarrow \theta - \eta \nabla_\theta \mathcal{L}$
12:    **end for**
13: **end for**

---

# B. Additional Experiments

### B.1. Ablations with Consistent Model Backbone

In Section 5, we compare QuRe with existing baselines on FashionIQ, CIRR, HP-FashionIQ, and CIRCO. However, model backbones are not unified, as we follow the comparison settings used in prior CIR methods (Baldrati et al., 2023; Liu et al., 2024; Bai et al., 2024). To ensure a fairer comparison, we conduct experiments using both BLIP and BLIP-2 backbones. Specifically, we train QuRe with the BLIP backbone to compare against Bi-BLIP4CIR. We also identify CoVR-2 (Ventura et al., 2024a), the latest version of CoVR-BLIP that adopts BLIP-2, and compare it with our original QuRe model.

Table 6, Table 7, and Table 8 present results on the CIRR, FashionIQ, HP-FashionIQ, and CIRCO datasets. QuRe with a BLIP backbone consistently outperforms Bi-BLIP4CIR. Moreover, CoVR-2, which adopts a BLIP-2 backbone, still underperforms compared to our original QuRe model with the same backbone.

*Table 6.* Performance comparison on the CIRR test dataset with consistent model backbones, where Recall$_s$@K represents Recallsubset@K. The best results are highlighted in bold.

| Method | Backbone | Recall@K | | | | Recall$_S$@K | | | Average |
|---|---|---|---|---|---|---|---|---|---|
| | | K=1 | K=5 | K=10 | K=50 | K=1 | K=2 | K=3 | R@5 + R$_S$@1 |
| Bi-BLIP4CIR | BLIP | 32.55 | 64.36 | 76.53 | 91.61 | 63.54 | 82.46 | 92.48 | 63.95 |
| **QuRe** | **BLIP** | **51.52** | **80.29** | **88.89** | **97.74** | **78.02** | **91.23** | **96.55** | **79.16** |
| CoVR-2 | BLIP-2 | 42.80 | 74.60 | 83.90 | 96.22 | 69.49 | 86.22 | 93.98 | 72.05 |
| **QuRe** | **BLIP-2** | **52.22** | **82.53** | **90.31** | **98.17** | **78.51** | **91.28** | **96.48** | **80.52** |

### B.2. Visualization of Score Steepness

QuRe defines a query-specific hard negative set by excluding false and easy negatives, aiming to enhance training effectiveness. Specifically, it selects images between the two largest drops in relevance scores following the target image. To analyze this steepness, we visualize the sorted relevance scores.

Figure 7 illustrates the relevance scores for a sample query from the FashionIQ dataset at two stages: before training and after the warm-up phase. QuRe identifies hard negatives as the images between the red and green lines, determined by the top two largest drops in relevance scores after the target. These steep declines-visible immediately before the red line and after the green line-suggest substantial drops in relevance (Xia et al., 2024), effectively separating false negatives, hard negatives, and easy negatives.

*Table 7.* Performance comparison on the FashionIQ validation dataset with consistent model backbones. The best results are highlighted in bold.

| Method | Backbone | Dress | | Shirt | | Toptee | | Average | | |
|---|---|---|---|---|---|---|---|---|---|---|
| | | R@10 | R@50 | R@10 | R@50 | R@10 | R@50 | R@10 | R@50 | Avg. |
| Bi-BLIP4CIR | BLIP | 39.12 | 62.92 | 39.21 | 62.81 | 44.37 | 67.06 | 40.90 | 64.26 | 52.58 |
| QuRe | **BLIP** | **40.80** | **64.90** | **45.93** | **65.90** | **52.07** | **72.87** | **46.27** | **67.89** | **57.08** |
| CoVR-2 | BLIP-2 | 46.41 | 69.51 | 49.75 | 67.76 | 51.86 | 72.46 | 49.34 | 69.91 | 59.63 |
| QuRe | **BLIP-2** | **46.80** | **69.81** | **53.53** | **72.87** | **57.47** | **77.77** | **52.60** | **73.48** | **63.04** |

*Table 8.* Comparison of HP-FashionIQ Preference Rate and CIRCO zero-shot performance with consistent model backbones. The best results are highlighted in bold.

| HP-FashionIQ | | | CIRCO (Zero-shot) | | | | | |
|---|---|---|---|---|---|---|---|---|
| Method | Backbone | Preference Rate (%) | Method | Backbone | m@5 | m@10 | m@25 | m@50 |
| Bi-BLIP4CIR | BLIP | 67.33 | Bi-BLIP4CIR | BLIP | 4.74 | 4.97 | 5.69 | 6.1 |
| QuRe | **BLIP** | **75.28** | QuRe | **BLIP** | **20.85** | **21.48** | **23.35** | **24.31** |
| CoVR-2 | BLIP-2 | 71.99 | CoVR-2 | BLIP-2 | 23.18 | 23.59 | 25.57 | 26.49 |
| QuRe | **BLIP-2** | **74.55** | QuRe | **BLIP-2** | **23.22** | **24.23** | **26.26** | **27.24** |

Since Figure 7 shows only a single query, we also present aggregated results across all queries in Figure 8. This aggregated visualization shows that, after the warm-up phase, hard negatives shift toward higher-ranked positions, likely capturing more true hard negatives. This supports our design choice of introducing a warm-up stage prior to hard negative selection. Without this stage, the selected negatives tend to include many easy examples, as observed in the left panel of the figure.

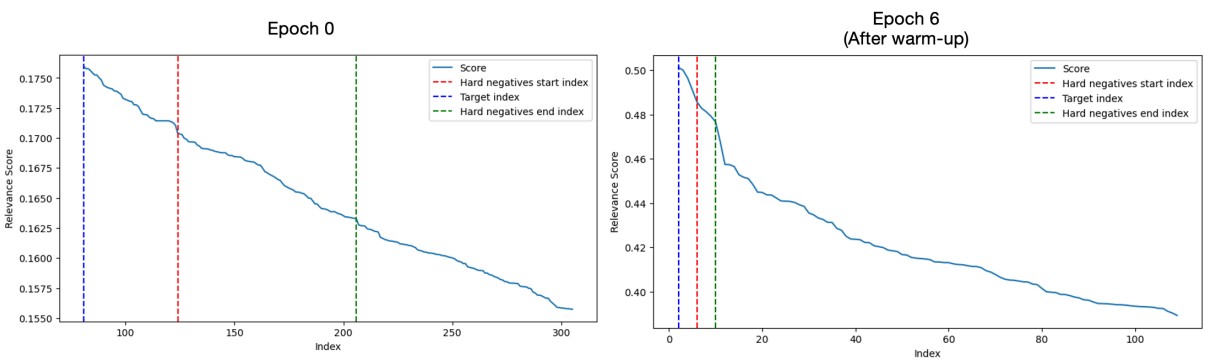

*Figure 7.* Visualization of sorted relevance scores for a FashionIQ query at two stages: prior to training and after completing the warm-up phase.

## C. HP-FashionIQ Dataset

### C.1. Data Annotation Examples

We conducted a data collection via Google Forms. Each form consisted of instructions and 25 questions, with each question including a query and two different sets of retrieved images. Each participant completed two forms, covering 50 queries from the FashionIQ validation dataset, with no overlap between participants.

Aggregated Sorted Score Plot after the Target

*Figure 8.* Visualization of sorted relevance scores aggregated over all FashionIQ queries at two stages: before training and after the warm-up phase.

Fig 9 shows the guidelines provided to participants, who were asked to score the retrieved image sets based on relevance to the query using a 5-point Likert scale. We gave participants a relatable scenario that required them to evaluate the results from two different online shopping malls based on their input. Figure 10 illustrates an example of a reference image (original image), relative text (user text), and two sets of retrieved images from different CIR models (Shopping Mall 1 and 2). Participants (1) rated the relevance of each set, (2) indicated which set they preferred, and (3) noted whether any results were irrelevant to the reference image or text, as shown in Figure 11.

*Figure 9.* Guidelines for user survey.

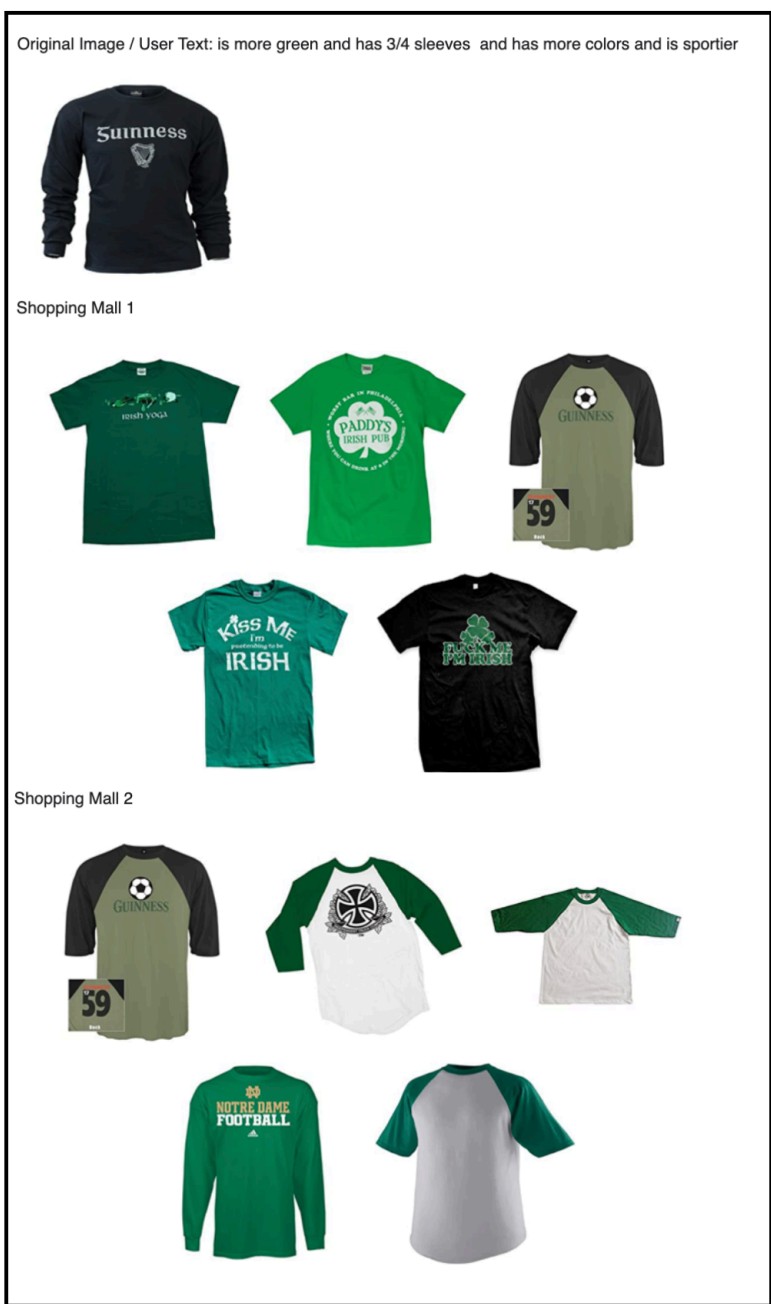

*Figure 10.* Example of query and two set of retrieved images in user survey.

*Figure 11.* Example of questions in user survey.

