# OpenReview forum: "QuRe: Query-Relevant Retrieval through Hard Negative Sampling in Composed Image Retrieval"
_ICML.cc/2025/Conference — ICML 2025 poster_

### Official Review · Reviewer_c7Tq · 2025-03-13

**Overall Recommendation:** 4

**Summary:**

This paper proposes a QURE method to retrieval the target image and mitigate the false negatives for the task of Composed Image Retrieval (CIR). The authors introduce a hard negative sampling strategy that select images positioned between two sharp relevance score drops after the target to filter false negative. The paper also introduces human-annotated HP-FashionIQ dataset that explicitly captures user preferences beyond target retrieval. Experiments on the FashionIQ, CIRR and CIRCO datasets validate the effectiveness of the proposed method.

**Claims And Evidence:**

The article is well written, there are no typos or spelling errors, and the arguments are well stated. The state-of-the-art and related topics are comprehensive. However, it is suggested to supplement some improved experiments about false negative results to facilitate readers' understanding.

**Essential References Not Discussed:**

The paper is well-written and fully cited.

**Experimental Designs Or Analyses:**

(1) Tables 2 and 3 compare different methods to the paper result (QURE). As described in Section 5, the authors use the BLIP-2 for the CIR task, but not all of the methods shown in table use BLIP-2. This makes it questionable whether much of the improvement in model performance is due to the introduction of BLIP-2.

(2) The authors show in Figure 6 the selection of the QURE method for hard-negative samples, but there is no report on the improvement of false negative results.

**Methods And Evaluation Criteria:**

The idea of QURE is novel and the HP-FashionIQ dataset provides a new evaluation metric for CIR. This contribution looks like alleviating the problem of false negative results in CIR research.

**Other Comments Or Suggestions:**

The example provided in figure 1 is helpful, but it would be beneficial to highlight the target image and the novelty of method for clarity. This is important for readers who may not be familiar with CIR.

**Other Strengths And Weaknesses:**

Strengths:

1. The idea of a hard negative sampling strategy is good.
2. The creation of the HP-FashionIQ dataset is a valuable contribution to the CIR field. It addresses the core limitations of current CIR evaluation metrics by focusing on user preferences rather than just retrieving target images.
3. The paper shows SoTA performance on the main CIR datasets.

Weaknesses:
1. The paper does not sufficiently report the improvement of the false negative of the proposed approach.

**Questions For Authors:**

The paper uses candidate images between two correlation sharp drops as hard negative samples is novel, but I am confused about how to determine whether it is a sharp drop or not. In addition, whether there will be some queries with multiple sharp drops affects the selection of hard negative samples.

**Relation To Broader Scientific Literature:**

The paper alleviates the problem of false negative results in the CIR dataset (FashionIQ) with a hard negative sampling strategy, which is one of the research directions in CIR.

**Theoretical Claims:**

I'm not sure if the hard-negative sampling strategy can be used to filter out false negatives for all queries. There may be cases where there are no hard-negative samples?

---

> ### Author Rebuttal · Authors · 2025-03-31
>
> We sincerely thank Reviewer c7Tq for the positive feedback and for recognizing the novelty of our work in both the proposed method and the dataset.
>
> **Q1 : There may be cases where there are no hard-negative samples?**
>
>
> It is true that the number of hard negatives can vary significantly depending on the query (e.g., if the user asks for a "blue short-sleeve t-shirt," then "blue long-sleeve shirts" might be considered hard negatives) and the available candidate images in the corpus (e.g., if there are no blue shirts, no hard negatives might exist). However, identifying true hard negatives for all queries is impractical in both real-world applications and existing CIR datasets. Instead, we empirically demonstrate that QuRe improves model performance by selecting **relatively** hard negatives from the corpus.
>
>
> **Q2 : Ablation studies for the BLIP-2 model architecture**
>
>
> We conducted additional experiments using both BLIP and BLIP-2 architectures to ensure a fair comparison between QuRe and other baselines. Specifically, we trained QuRe on the BLIP backbone to compare with Bi-BLIP4CIR. For CoVR, we identified its latest version, CoVR-2[r1], which adopts BLIP-2 as its backbone, and compared it with our original QuRe model. Notably, the current state-of-the-art method SPRC originally used BLIP-2, and QuRe already outperformed it in our original submission. The results are presented across the following datasets:
>
>
> - **CIRR**: QuRe outperforms Bi-BLIP4CIR and CoVR-2 with average Recall improvements of **11.53** and **6.04**, respectively.
> - **FashionIQ**: QuRe outperforms Bi-BLIP4CIR and CoVR-2 with average Recall improvements of **4.50** and **3.41**, respectively.
> - **CIRCO**: QuRe outperforms Bi-BLIP4CIR and CoVR-2 with average mAP improvements of **17.12** and **0.53**, respectively.
> - **HP-FashionIQ**: QuRe outperforms Bi-BLIP4CIR and CoVR-2 with higher human preference rates of **7.95** and **2.56**, respectively.
>
>
> Please refer to our response in $\text{\color{blue}\textbf{Q2 under Reviewer 6GLy}}$ for detailed results.
>
>
>
> **Q3 : Report on the improvement of false negative results**
>
>
> We agree that providing additional qualitative results, similar to $\text{\color{red}\textbf{Figure 6}}$, helps illustrate the improvements in handling false negatives. Using the given query, we extracted the top 4 retrieved results from Bi-BLIP4CIR, CoVR-BLIP, SPRC, and our method, QuRe. The results are shown in the following figure:
>
>
> ### **<https://anonymous.4open.science/r/QuRe-ICML-Rebuttal/QuRe_FalseNegatives.png>**
>
> As shown, QuRe successfully retrieves relevant t-shirts that match the query attributes, such as navy blue color (close to black, like the query image) and the presence of a chicken or bird in the center. In contrast, other baselines retrieve less relevant images, such as those in light blue or without the chicken/bird. Such irrelevant results can lead to user dissatisfaction, as user satisfaction is generally proportional to the number of relevant items in the retrieved set. These results highlight the effectiveness of QuRe in assigning higher scores to relevant images by explicitly addressing false and easy negatives during training.
>
> **Q4 : Add target image and novelty of the method in Figure 1**
>
>
> Thank you for pointing this out. We agree that highlighting the target image for each query in $\text{\color{red}\textbf{Figure 1}}$ would improve clarity and overall understanding. Additionally, we will include the $\text{\color{blue}\textbf{figure provided in Q3}}$ ([**QuRe_FalseNegatives.png**](https://anonymous.4open.science/r/QuRe-ICML-Rebuttal/QuRe_FalseNegatives.png)) in the appendix to illustrate the effectiveness of our method, QuRe, compared with existing baselines.
>
>
> **Q5 : How is sharpness defined, and what if multiple sharp drops exist?**
>
>
> To determine whether the drop is sharp, we compute the relevance scores of all candidate images for each query. We then sort the scores and identify the top-2 largest drops occurring after the target. The corresponding images are selected as the hard negative set. An example of these steep degradation patterns is shown in $\text{\color{blue}\textbf{Q2 under Reviewer 13Df}}$ ([**QuRe_steep_one.png**](https://anonymous.4open.science/r/QuRe-ICML-Rebuttal/QuRe_steep_one.png)). While there may be multiple sharp drops, we believe that the top-2 degradations after the target likely represent the boundaries between false negatives, hard negatives, and easy negatives. We agree that more advanced approaches to identifying such transitions could be explored in the future. We will include this as a discussion point in the final version of the paper.
>
>
> We will update the manuscript to incorporate the feedback discussed in this rebuttal. We sincerely hope that our responses, along with the originality of our contributions, have addressed your concerns. If any questions remain, we would be glad to offer further clarification.

---

> > ### Comment · Reviewer_c7Tq · 2025-04-06
> >
> > I appreciate the rebuttal from the authors, but it does not address my fourth question very well, which is about the novelty of the paper.  I'm changing my final rating to "weak accept".

---

> > > ### Author Response · Authors · 2025-04-07
> > >
> > > Thank you for your clarification. We first apologize; there might have been a misunderstanding regarding your fourth question on the novelty of our work. To better illustrate both the CIR task and the novelty of our method, QuRe, we have re-drawn Figure 1:
> > >
> > >
> > > ### **<https://anonymous.4open.science/r/QuRe-ICML-Rebuttal/QuRe_Figure_1.png>**
> > >
> > >
> > > In this revised figure, we retain the original CIR example, **while more clearly emphasizing QuRe’s novelty over existing CIR methods**. Specifically, QuRe sorts the candidate images by their relevance scores to the query and defines the hard negative set as the images between the two largest drops after the target. This approach excludes both false and easy negatives, unlike existing methods that rely on fixed candidate pools.
> > >
> > >
> > > We also annotate the **target image** and highlight that while existing methods may retrieve the target, they often include irrelevant images. In contrast, QuRe retrieves both the target and other relevant images.
> > >
> > >
> > > We appreciate your feedback, which has helped us improve the clarity of our contributions. We will revise the manuscript accordingly and sincerely hope this addresses your concern. We would be grateful if you would consider this clarification in your re-evaluation.

---

### Official Review · Reviewer_6GLy · 2025-03-13

**Overall Recommendation:** 3

**Summary:**

The paper introduces the QURE algorithm, leveraging the BLIP-2 framework and a Hard Negative Sampling strategy to address the challenges in Cross-Image Retrieval (CIR). The novel approach is demonstrated using a custom dataset, HP-FashionIQ. While the approach is innovative and effectively addresses key pain points, the paper does not provide ablation studies for the BLIP-2 model, making it difficult to discern the individual contributions of the model enhancements from the sampling strategy

**Claims And Evidence:**

The effectiveness of the Hard Negative Sampling strategy is well-supported by empirical evidence. However, the paper lacks a crucial ablation study of the BLIP-2 framework, which is necessary to isolate and understand its specific contributions relative to the overall performance improvements claimed.

**Essential References Not Discussed:**

The current literature review is sufficient but would be enhanced by including discussions on theoretical frameworks that specifically address the role and impact of relevance scores in negative sampling within similar contexts.

**Experimental Designs Or Analyses:**

The experimental design demonstrates a thorough understanding of the practical applications but lacks detailed ablation studies on the BLIP-2 model. This omission is critical as it hinders the clear differentiation of the effects of the model's capabilities from those of the sampling strategy.

**Methods And Evaluation Criteria:**

The QURE algorithm and Hard Negative Sampling approach are clearly articulated and address the identified issues in CIR tasks effectively. The validation on the HP-FashionIQ dataset appropriately benchmarks the model's performance. Nevertheless, the absence of BLIP-2 specific ablation studies undermines the ability to fully evaluate the method's effectiveness.

**Other Comments Or Suggestions:**

It is recommended that the authors include ablation studies for the BLIP-2 model to clarify its contributions and provide a detailed mathematical justification for the relevance-based hard negative selection process.

**Other Strengths And Weaknesses:**

Strengths:
1.the paper is well-written, and easy to understand.
2.It effectively tackles key pain points in Cross-Image Retrieval tasks, notably through its novel Hard Negative Sampling strategy which improves model robustness
3.The detailed analysis and clear visualizations of the Hard Negative Sampling strategy aid in understanding and highlight its practical impact.
Weaknesses:
1.the lack of ablation studies for the BLIP-2 framework
2.insufficient theoretical justification for the relevance score methodology in Hard Negative Sampling are significant drawbacks

**Questions For Authors:**

No

**Relation To Broader Scientific Literature:**

The paper positions itself well within the existing literature and effectively addresses significant challenges in the field. It brings innovative solutions to the forefront, though it could benefit from a more detailed discussion on theoretical frameworks related to negative sampling.

**Theoretical Claims:**

While practical in nature, the paper does not adequately justify the theoretical basis for the Hard Negative Sampling strategy's relevance score methodology. A rigorous mathematical or logical explanation is required to substantiate the claims made about the selection of negatives based on their relevance scores.

---

> ### Author Rebuttal · Authors · 2025-03-31
>
> We sincerely thank Reviewer 6GLy for recognizing the contributions of our work and for providing valuable suggestions.
>
>
> **Q1 : Mathematical justification for the Hard Negative Sampling strategy**
>
>
> We re-emphasize an important limitation in CIR datasets. Since only one or a few target images are annotated per query, **it is not feasible to accurately determine which images are false, hard, or easy negatives**. Hence, we first provide a theoretical justification for our approach, which is why defining the hard negative set and sampling from it in each epoch is effective in our scenario. The justification shows sampling negative from the hard negative set yields a higher expected loss than sampling from all corpus, thereby guiding the model to focus more on the defined hard negatives during training. Please refer to the following link:
>
>
> ### **<https://anonymous.4open.science/r/QuRe-ICML-Rebuttal/QuRe_Justification.pdf>**
>
>
> However, if the defined hard negative set is improperly constructed and includes false negatives, it can misguide training. In $\text{\color{red}Figure 4}$, we observe a clear performance improvement immediately after introducing the hard negative set, compared to sampling from the full corpus. Conversely, sampling from the Top-K set, which contains more false negatives than our defined set, leads to performance degradation.
>
>
> **Q2 : Ablation studies for the BLIP-2 model architecture**
>
>
> We conducted additional experiments using both BLIP and BLIP-2 model structures. Specifically, we trained QuRe with the BLIP backbone to compare with Bi-BLIP4CIR. We also identified the latest version of CoVR-BLIP, CoVR-2[r1], which uses BLIP-2, and compared it with our original QuRe model. We report results on the CIRR, FashionIQ, HP-FashionIQ, and CIRCO datasets. Notably, the current state-of-the-art method SPRC originally used BLIP-2, and QuRe already outperformed it in our original submission.
>
>
> | CIRR | backbone | Recall@1 | Recall@5 | Recall@10 | Recall@50 | Recall s@1 | Recall s@2 | Recall s@3 | Mean 5 + 1 |
> |:------------:|:--------:|:--------:|:--------:|:---------:|:---------:|:----------:|:----------:|:----------:|:----------:|
> | Bi-BLIP4CIR  | BLIP     | 32.55    | 64.36    | 76.53     | 91.61     | 63.54      | 82.46      | 92.48      | 63.95      |
> | **QuRe**     | **BLIP** | **51.52**| **80.29**| **88.89** | **97.74** | **78.02**  | **91.23**  | **96.55**  | **79.16**  |
> | CoVR-2       | BLIP-2   | 42.80    | 74.60    | 83.90     | 96.22     | 69.49      | 86.22      | 93.98      | 72.05      |
> | **QuRe**     | **BLIP-2**| **52.22**| **82.53**| **90.31** | **98.17** | **78.51**  | **91.28**  | **96.48**  | **80.52**  |
>
>
>
>
> | **FashionIQ** | **backbone** | **Dress - Recall@10** | **Dress - Recall@50** | **Shirt - Recall@10** | **Shirt - Recall@50** | **TopTee - Recall@10** | **TopTee - Recall@50** | **Recall@10** | **Recall@50** | **Mean** |
> |:---------------------:|:-------------------------:|:---------------------:|:---------------------:|:---------------------:|:---------------------:|:----------------------:|:----------------------:|:-------------:|:-------------:|:---------:|
> | Bi-BLIP4CIR | BLIP | 39.12 | 62.92 | 39.21 | 62.81 | 44.37 | 67.06 | 40.90 | 64.26 | 52.58 |
> | **QuRe** | **BLIP** | **40.80** | **64.90** | **45.93** | **65.90** | **52.07** | **72.87** | **46.27** | **67.89** | **57.08** |
> | CoVR-2 | BLIP-2 | 46.41 | 69.51 | 49.75 | 67.76 | 51.86 | 72.46 | 49.34 | 69.91 | 59.63 |
> | **QuRe** | **BLIP-2** | **46.80** | **69.81** | **53.53** | **72.87** | **57.47** | **77.77** | **52.60** | **73.48** | **63.04** |
>
>
> | **CIRCO** | **backbone** | **mAP@5** | **mAP@10** | **mAP@25** | **mAP@50** |
> |-------------------|:-------------------------:|-----------|------------|------------|------------|
> | Bi-BLIP4CIR | BLIP | 4.74 | 4.97 | 5.69 | 6.1 |
> | **QuRe** | **BLIP** | **20.85** | **21.48** | **23.35** | **24.31** |
> | CoVR-2 | BLIP-2 | 23.18 | 23.59 | 25.57 | 26.49 |
> | **QuRe** | **BLIP-2** | **23.22** | **24.23** | **26.26** | **27.24** |
>
>
>
> | **HP-FashionIQ** | **backbone** | **Preference Rate (%)** |
> |--------------------------|:-------------------------:|-------------------------|
> | Bi-BLIP4CIR | BLIP | 67.33 |
> | **QuRe** | **BLIP** | **75.28** |
> | CoVR-2 | BLIP-2 | 71.99 |
> | **QuRe** | **BLIP-2** | **74.55** |
>
>
>
> The results show that QuRe with a BLIP backbone consistently outperforms Bi-BLIP4CIR. CoVR-2, which uses a BLIP-2 backbone, still underperforms our original QuRe with BLIP-2.
>
>
> [r1] Ventura, Lucas, et al. "CoVR-2: Automatic Data Construction for Composed Video Retrieval." IEEE Transactions on Pattern Analysis and Machine Intelligence (2024).
>
>
> We will revise the manuscript to incorporate the points addressed in this rebuttal. We hope our responses and the novelty of our contributions have sufficiently addressed your concerns. If there are any remaining issues or points that need further clarification, we would be more than happy to provide additional details.

---

### Official Review · Reviewer_13Df · 2025-03-14

**Overall Recommendation:** 3

**Summary:**

This work introduces a new method of QuRe to tackle the problem of composed image retrieval. The proposed method adopts and tailors the hard negative mining to emphasize not only the ranking of the target image, but also other relevant images in the dataset, aiming at improving the overall recall. Experiments on benchmark dataset demonstrate improved performance of the proposed method.

**Claims And Evidence:**

The claims could be problematic. See weakness.

**Essential References Not Discussed:**

The related works are essential.

**Experimental Designs Or Analyses:**

The experimental designs could be biased. See weakness.

**Methods And Evaluation Criteria:**

The method makes sense.

**Other Comments Or Suggestions:**

Another strategy to validate the effectiveness of the proposed motivation is to apply the proposed method on a wide range of the datasets beyond the HP-FashionIQ and CIRR. Is the motivation to take advantage of the inherent bias of above datasets?

**Other Strengths And Weaknesses:**

The motivation of the proposed method could be problematic. The proposed method challenges the data labeling of existing benchmark datasets – existing benchmark datasets can mislabel relevant images as the false negative ones and thus could be highly biased. In this case, detecting and prioritizing hard negatives could be beneficial.  Yet, there lacks justification and thorough study to this motivation.

It seems that no visualizations and discussions on the steeps during the optimization and for different datasets.

**Questions For Authors:**

The proposed method remains unclear to me. The proposed  method monitors the steeps to distinguish between different levels of the negatives. Yet, how to make sure the quality of the ranking and steeps during optimization, especially at the initial stage? More illustrations or empirical evidence expected.

**Relation To Broader Scientific Literature:**

Could be benificial to multimodal learning.

**Theoretical Claims:**

No proofs needed.

---

> ### Author Rebuttal · Authors · 2025-03-31
>
> We sincerely thank Reviewer 13Df for the insightful comments and the time dedicated to reviewing our manuscript. Below, we provide detailed responses to each of your points.
>
> **Q1 : Justification and thorough study to the motivation of the proposed method**
>
> The primary motivation behind our method stems from a key limitation of existing CIR datasets, where only one or a few target images are annotated per query. As a result, false negatives are often included as negatives during training. Existing baselines typically treat all non-target images as negatives, which delays convergence and degrades performance [r1].
>
>
> Our method addresses this issue by modifying the contrastive learning objective to compare the true positive against a single negative with increasing difficulty. This is achieved by selecting a small set of hard negatives for each query. We validate the effectiveness of our approach by achieving state-of-the-art performance on both CIRR and FashionIQ, and demonstrating the highest correlation with human preferences on HP-FashionIQ.
>
>
> We clarify the motivation of our study, we added visualizations of the steep drops in relevance scores (see $\text{\color{blue}\textbf{Q2}}$) and provided a mathematical justification in response to $\text{\color{blue}\textbf{Q1 in Reviewer 6GLy}}$.
>
>
> [r1] Huynh, Tri, et al. "Boosting contrastive self-supervised learning with false negative cancellation." Proceedings of the IEEE/CVF winter conference on applications of computer vision. 2022.
>
>
> **Q2 : Visualizations and discussions on the steep**
>
>
> Thank you for pointing this out. We agree that visualizing the steeps during optimization is beneficial. To illustrate this, we plotted the relevance scores for one sample from the FashionIQ dataset at two stages: before training and after the warm-up training. Please refer to the following link:
>
>
> ### **<https://anonymous.4open.science/r/QuRe-ICML-Rebuttal/QuRe_steep_one.png>**
>
>
> We defined the hard negatives as the images between the red and green lines. By selecting the top-2 largest drops in relevance score following the target, we observed steep degradations immediately before the red line and after the green line. These steep drops indicate substantial decreases in relevance[r2], suggesting that the selected boundaries effectively separate false negatives, hard negatives, and easy negatives.
>
> As the above figure shows only a single query, we also aggregated the results across all queries, which can be found here:
>
>
> ### **<https://anonymous.4open.science/r/QuRe-ICML-Rebuttal/QuRe_steep_agg.png>**
>
>
> This aggregated visualization demonstrates that after the warm-up phase, the hard negatives shift toward higher ranks, likely capturing more true hard negatives. This supports our design choice of including a warm-up stage before defining hard negatives. Without a warm-up, the hard negative set would contain many easy negatives, as shown in the left figure.
>
>
> [r2] Xia, Peng, et al. "Mmed-rag: Versatile multimodal rag system for medical vision language models." ICLR 2025.
>
>
> **Q3 : Apply QuRe on a wide range of the datasets beyond the HP-FashionIQ and CIRR**
>
>
> We evaluated our method on four datasets: CIRR, FashionIQ, HP-FashionIQ, and CIRCO. CIRR and FashionIQ are two representative datasets in CIR, where CIRR covers general domains (e.g., people, animals, food) and FashionIQ focuses on fashion-related queries (e.g., shirt, dress, etc). We agree that evaluating on a broader range of datasets would further validate our approach. To this end, we created the HP-FashionIQ dataset to better capture human preferences. We additionally evaluated on CIRCO, a dataset derived from COCO, which enables relevance-based retrieval evaluation using mAP in a zero-shot setting.
>
>
> **Q4 : How to make sure the quality of the ranking and steeps especially at the initial stage?**
>
>
> If the model is not sufficiently trained and the relevance scores are not reliable, our proposed method, which defines the hard negative set based on relevance scores, might produce inaccurate results. To address this, we include a warm-up phase where the model is trained by sampling negatives from the entire corpus without defining a hard negative set. As shown in $\text{\color{red}\textbf{Figure 4}}$, the model achieves comparable performance even when negatives are sampled from the full corpus ("All corpus"). Additionally, the **QuRe_steep_agg.png** visualization in $\text{\color{blue}\textbf{Q2}}$ demonstrates that defining the hard negative set without warm-up training tends to select easy negatives, which undermines the effectiveness of hard negative mining.
>
> We will revise the manuscript to reflect the points raised in this rebuttal. We hope our responses and the novelty of our contributions have adequately addressed your concerns. Should any issues remain, we are happy to provide further clarification.

---

> > ### Comment · Reviewer_13Df · 2025-04-07
> >
> > I appreciate the authors' effort in rebuttal. Some of my concerns are solved, including visualizations and illustrations on datasets. Yet I'm not sure about the generalizability of the proposed method to a more broader spectrum of datasets. I'm willing to raise my score.

---

### Decision · Program_Chairs · 2025-05-01

**Decision:**

Accept (poster)

**Comment:**

The paper proposes a retrieval framework that improves composed image retrieval by introducing a hard negative sampling strategy based on steep drops in relevance scores. It also presents HP-FashionIQ, a new benchmark for evaluating alignment with human satisfaction. The reviewers find the motivation clear and the method well-executed, with good empirical performance across multiple CIR datasets. While some concerns are raised regarding in-depth discussion, theoretical basis, architectural ablations, etc, the authors provide additional experiments and visualizations that help clarify these aspects. Several questions remain regarding the technical novelty and the generalizability of the method, as the focus is primarily on fashion items. Overall, the paper offers certain contribution to the CIR community.